# SADT: Combining Sharpness-Aware Minimization with Self-Distillation for Improved Model Generalization

**Masud An-Nur Islam Fahim**
University of Vaasa
Vaasa, Finland
`masud.fahim@uwasa.fi`

**Jani Boutellier**
University of Vaasa
Vaasa, Finland
`jani.boutellier@uwasa.fi`

## Abstract

Methods for improving deep neural network training times and model generalizability consist of various data augmentation, regularization, and optimization approaches, which tend to be sensitive to hyperparameter settings and make reproducibility more challenging. This work jointly considers two recent training strategies that address model generalizability: sharpness-aware minimization, and self-distillation, and proposes the novel training strategy of Sharpness-Aware Distilled Teachers (SADT). The experimental section of this work shows that SADT consistently outperforms previously published training strategies in model convergence time, test-time performance, and model generalizability over various neural architectures, datasets, and hyperparameter settings.

## 1 Introduction

Over the recent years that machine learning has rapidly developed, researchers have discovered a variety of data pre-processing steps and training strategies to speed up the training process and/or achieve better results. The spectrum of adopted approaches includes data augmentation, regularization and hyperparameter tuning methods. As a by-product of these numerous training related alternatives, the reproducibility and instability of model training has become a challenge.

Recently, *sharpness-aware* approaches [1–4] have addressed the training instability issue by focusing on training loss fluctuation around the bounded neighborhood of the model parameters for improving model generalizability. In essence, sharpness-aware minimization (SAM) approaches probe the linearly dependent subset of the current parameter space to tune given input images and reach wider minima by better regularization. In contrast, *self-distillation* approaches [5, 6] improve model performance by training a better-performing student model based on a previously trained teacher model. Surprisingly, when this process is repeated over multiple rounds, the performance of the student model improves, despite that no new data is provided to the student models [7]. Researchers argue that the self-distillation process is a progressive regularization method that needs to be studied further [8].

This work presents a study on the fundamental mechanisms of both sharpness-aware minimization and self-distillation methods, and consequently proposes a structured family of training strategies to improve training performance. The proposed Sharpness-Aware Distilled Teachers (SADT) approach creates an improved variant of the teacher model from the original teacher model within a single distillation round. Consequently, we show that SADT achieves considerable improvement in convergence speed and generalizability over other works [1, 9, 10] that operate in a single training round.

The contributions of this paper are:

Has it Trained Yet? Workshop at the Conference on Neural Information Processing Systems (NeurIPS 2022).

- We propose SADT, a novel family of training strategies that combines sharpness-aware minimization with self-distillation.
- Experimental results, which show that SADT is less sensitive to training parameter settings than several other related methods [1, 9–11], and provides consistently better results.

## 2 Related Work

The generalizability of a deep neural network training process depends on multiple factors: the adopted data augmentation, gradient update and regularization policies, as well as the network itself.

*Data augmentation* approaches (e.g., [12]) strengthen the training procedure by preventing overfitting, increasing feature diversity, and by promoting saliency-aware learning. Beyond basic approaches that adopt geometric and spatial transformations, more advanced schemes have been proposed: CutMix [11] randomly mixes image patches, CutOut [13] introduces regional dropout in rectangular forms, MixUp [14] performs regional blending, PuzzleMix [15] addresses adversarial attacks, whereas SaliencyMix [12] is a refined version of CutMix [11], focusing on salient patches instead of random patches.

*Regularization* methods operate around parameter space perturbation [16–18], gradient update [9, 10, 1–4], and normalization [19–22] strategies. Methods that focus on gradient update policies, concentrate on gradient behavior and according changes [9, 10]. Sharpness-aware methods [1–4] can be considered as a branch of gradient manipulation approaches, offering improved generalizability. Finally, perturbation schemes [16–18] work on the feature space, parameters, and gradients in order to regularize [19–22] the neural network.

*Distillation* approaches improve model generalizability by transferring knowledge from a teacher network to a compact student network. Typically, knowledge extraction from the teacher model is done by means of soft labels [23–26], intermediate layer output [27], or feature maps [28]. Self-distillation approaches, on the other hand, use identical teacher and student networks and introduce model training self-guidance by means of data augmentation [29, 30], feature refinement [6], or use of auxiliary classifiers [6, 5].

## 3 Proposed Method

Sharpness-aware minimization and self-distillation procedures to some extent contain intrinsic similarities. Conceptually, sharpness aware minimization [1–4] approaches seek wider minima in the loss surface, while performing optimization over the given objective. More formally, let $\mathcal{S}_{\text{train}} = \{x_i, y_i\}_{i=1}^n$ be the training dataset and $\ell_i(w)$ be the cost of the model parameterized by weights $w \in \mathbb{R}^{|w|}$, evaluated at any given point $(x_i, y_i)$. For a given perturbation component $\delta$, $\ell_i(w') = \ell_i(w + \delta)$ is the perturbed cost. Then, the sharpness related to a set of points $\mathcal{S} \subseteq \mathcal{S}_{\text{train}}$ is defined as [31]:

$$s(w, \mathcal{S}) \triangleq \max_{\|\delta\|_2 \leq \rho} \frac{1}{|\mathcal{S}|} \sum_{i:(x_i, y_i) \in \mathcal{S}} \ell_i(w + \delta) - \ell_i(w)$$

where $\rho$ [1] is the neighbourhood size. Whereas the older works in the field generally define sharpness as $\mathcal{S} = \mathcal{S}_{\text{train}}$, the recent work [31] defines it as the average of the all batches from $\mathcal{S}_{\text{train}}$. It can be seen that the sharpness term seeks to minimize the *divergence* between the original $\ell_i(w)$ and the perturbed model $\ell_i(w')$ with respect to the hard labels [31]. Similarly as in sharpness aware minimization above, the concept of divergence is also present in self-distillation:

**Definition 1.** For the models $f(.; w)$ and $f(.; w')$ in the same weight space $\mathbb{R}^{|w|}$, given $x_i \in \mathcal{S}$, the generalizability gap between $w$ and $w'$ is expressed as the Kullback-Leibler divergence,

$$d_p(w, w') = \mathbb{E}_x \left[ D_{\text{KL}}(f(x; w) \| f(x; w')) \right] \text{ [32]}$$

where $D_{\text{KL}}(\cdot \| \cdot)$ denotes the KL divergence and $d_p(w, w')$ the model divergence between $w$ and $w'$.

Assuming $d_p(w, w') \geq 0$, self-distillation approaches try to minimize this distance by matching the logits from $f(.; w)$ and $f(.; w')$. The more the divergence $d_p(w, w')$ reduces, the more regularized the model becomes, and the possibility of reaching wider minima increases. Similarly, sharpness studies argue for better generalization through reducing the sharpness term over the training process. However, lower sharpness does not always guarantee better test time performance [31].

Sharpness-aware algorithms are also related to gradient perturbation; hence, larger batch sizes might impact generalizability. On the other hand, distillation approaches typically use multiple training rounds and require annealing parameters [7, 6].

In the following, we propose SADT: an approach for combining sharpness-aware and self-distillation schemes, with the aim of leveraging their advantages.

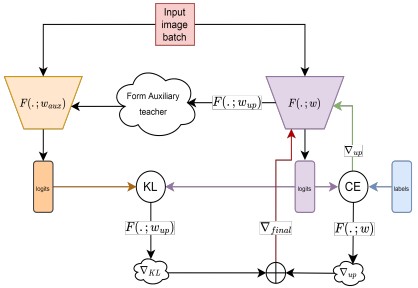

Figure 1: The general SADT flow chart. Green and red arrows indicate initial backpropagation, and final backpropagation, respectively. The cloud symbol covers the custom choice of noise aggregation to $f(.; w_{up})$. Estimated $\nabla_{KL}$ is obtained from $f(.; w_{up})$ instead of $f(.; w_{aux})$.

### 3.1 Sharpness-Aware Distilled Teachers

The main idea behind SADT is forming an *auxiliary teacher* by loosely adopting parameter space perturbation used by SAM [1], and consequent self-distillation, followed by gradient aggregation for final backpropagation, to improve training results. Figure 1 shows an overview of the general SADT flow, explained below in higher detail: we regard $w_{up}$ as the *updated self-teacher model*, the logits before the update step $f(x_i; w)$, and $\nabla_{up}$ as the gradient set used for computing $f(.; w_{up})$. An auxiliary self-teacher model $f(.; w_{aux})$ is created by adding random noise $\mathcal{N}(0, \sigma_w^2)$ to $f(.; w_{up})$ (Here, $\sigma_w^2$ is the standard deviation which equals the initial learning rate 0.0001). Next, $f(.; w_{aux})$ is used to infer the perturbed logits $f(x_i; w_{aux})$, which are used for soft-label matching similar to self-distillation studies.

For self-distillation, the Kullback-Leibler divergence $D_{KL}(f(x_i; w) \| f(x_i; w_{aux})$ is minimized by comparing soft labels between $f(x_i; w)$ and $f(x_i; w_{aux})$. In the final backward pass, we first compute $\nabla_{aux}$, followed by direct aggregation between $\nabla_{up}$ and $\nabla_{aux}$, resulting in $\nabla_{final}$. Then, the auxiliary teacher refers to $f(.; w_{up})$ by subtracting $\mathcal{N}(0, \sigma_w^2)$ from $f(.; w_{aux})$, and final backpropagation is performed using $f(.; w_{up})$ and $\nabla_{final}$.

Following this general procedure, the proposed SADT approach can be detailed to three variants:

**Variant 1.** The auxiliary teacher model $f(.; w_{aux})$ is formed by adding random noise to every layer of the self-teacher $f(.; w_{up})$. Soft label matching and final gradient descent operations are as described above.

**Variant 2.** Two auxiliary teachers $f(.; w_{aux1})$ and $f(.; w_{aux2})$ are introduced by adding noise to the final convolutional layer, and the final dense layer of $f(.; w_{up})$, respectively. Here, KL divergence between $[f(x_i; w_{aux1}), f(x_i; w)]$ and $[f(x_i; w_{aux2}), f(x_i; w)]$ is minimized, followed by aggregation between $\nabla_{aux1}$, $\nabla_{aux2}$ and $\nabla_{up}$ to obtain $\nabla_{final}$. $\nabla_{final}$ and $f(.; w_{up})$ are used in the final gradient descent.

**Variant 3.** Add noise to $\nabla_{up}$ to obtain $\nabla_{noisy}$, which then forms $f(.; w_{aux})$ by gradient ascent. After measuring KL divergence as in the general procedure, this version uses $\nabla_{up}$, $\nabla_{aux}$, and $f(.; w_{up})$ to perform final gradient descent.

In the following, we evaluate the these three variants of SADT against recent comparable training strategies.

## 4 Experiments

Below, the family of the proposed SADT variants is evaluated using multiple datasets and neural architectures. In particular, each training procedure has been performed from scratch, and independent of any pre-training steps. The source code is available at `https://github.com/DeepUVaasa/SADT`

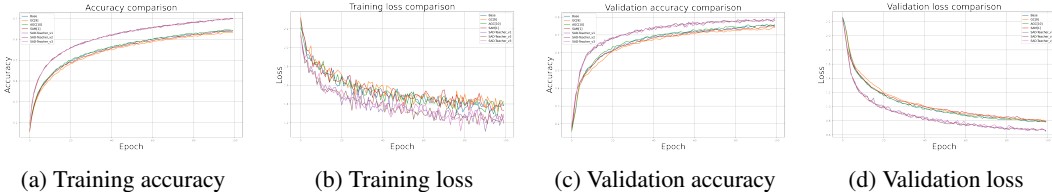

| (a) Training accuracy | (b) Training loss | (c) Validation accuracy | (d) Validation loss |

Figure 2: Training time comparison example: Simple CNN model, CIFAR10, batch size 2048.

## 4.1 Training setup details

We evaluate SADT by classification tasks using CIFAR10 and CIFAR100 datasets. The neural architectures used are Simple CNN (custom model of 3 conv and 3 dense layers), VGG [33], and InceptionResNet [34]. The SADT variants are compared against related works in model generalization: Gradient Centralization (GC) [9], Adaptive Gradient Clipping (AGC) [10], and Sharpness-Aware Minimization (SAM) [1]. In particular, self-distillation methods were not included to the comparison, as they require multiple training rounds, which makes them incompatible with our single-round experimental setting. In order to make the training landscape uniform for all methods considered, the training always starts from the same initial point, and all methods use the same optimizer (Adam), learning rate scheduler (cosine-decay with initial rate of 0.0001), batch size BS (512 and 2048), epoch count (200 for BS 512 and 370 for BS 2048), and data augmentation scheme (CutMix [11]).

## 4.2 Classification task results

Figure 2 depicts the training behavior of SADT against previous works: a clear performance gap between the SADT variants and other methods is visible in each performance metric. *Most importantly, SADT provides significantly faster training and higher accuracy compared to other approaches.* Table 1, on the other hand, presents test-time results showing best scores in boldface. *Baseline* refers to the original model amplified by CutMix [11], whereas the results in the rows below build on top of the baseline. Looking at the results of Table 1, Gradient Centralization [9] shows decreased performance with higher batch sizes. AGC [10] provides better performance than GC in all but one case, whereas SAM [1] outperforms the previous especially with the larger CIFAR100 dataset. Finally, the proposed SADT approach presents superior performance independent of batch size, model architecture, or dataset. In particular, the SADT test time scores with batch size 2048 almost equal baseline scores of batch size 512. Hence, SADT can offer a better alternative to compared training schemes while avoiding computational costs of repeated forward and backward passes. It is not possible to nominate a clear winner among SADT variants, hinting that the SADT could benefit from further study.

| Compared methods | Simple CNN | | VGG Net | | InceptionResNet | |
|---|---|---|---|---|---|---|
| | 512 | 2048 | 512 | 2048 | 512 | 2048 |
| Baseline | 0.783 | 0.745 | 0.841 | 0.817 | 0.817 | 0.775 |
| GC [9] | 0.784 | 0.738 | 0.843 | 0.804 | 0.801 | 0.763 |
| AGC [10] | 0.786 | 0.746 | 0.854 | 0.818 | 0.811 | 0.782 |
| SAM [1] | 0.784 | 0.730 | 0.856 | 0.826 | 0.824 | 0.766 |
| SADT Variant1 | 0.801 | 0.774 | 0.870 | 0.851 | 0.847 | 0.822 |
| SADT Variant2 | 0.809 | 0.775 | 0.876 | 0.847 | 0.847 | 0.811 |
| SADT Variant3 | 0.812 | 0.777 | 0.852 | 0.854 | 0.852 | 0.822 |

(a) CIFAR10

| Compared methods | Simple CNN | | VGG Net | | InceptionResNet | |
|---|---|---|---|---|---|---|
| | 512 | 2048 | 512 | 2048 | 512 | 2048 |
| Baseline | 0.381 | 0.344 | 0.517 | 0.485 | 0.521 | 0.466 |
| GC [9] | 0.370 | 0.310 | 0.533 | 0.475 | 0.515 | 0.462 |
| AGC [10] | 0.382 | 0.352 | 0.524 | 0.495 | 0.526 | 0.471 |
| SAM [1] | 0.385 | 0.363 | 0.542 | 0.498 | 0.525 | 0.479 |
| SADT Variant1 | 0.406 | 0.394 | 0.552 | 0.564 | 0.560 | 0.527 |
| SADT Variant2 | 0.425 | 0.394 | 0.565 | 0.565 | 0.553 | 0.513 |
| SADT Variant3 | 0.413 | 0.395 | 0.560 | 0.572 | 0.566 | 0.529 |

(b) CIFAR100

Table 1: Test time accuracy for all methods [1, 9–11] for CIFAR10 and CIFAR100 datasets.

## 5 Conclusion

This study presented the Sharpness-Aware Distilled Teachers training strategy where the given network is optimized using a combination of sharpness-aware minimization and self-distillation. The proposed method aggregates the gradients from different stages, which aids in improving the overall training process. Presented results on training and test time performance on two datasets and three neural architectures show that SADT provides faster convergence and consistently better results than previous works.

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
