# OpenReview forum: "SADT: Combining Sharpness-Aware Minimization with Self-Distillation for Improved Model Generalization"
_NeurIPS.cc/2022/Workshop/HITY — HITY Workshop NeurIPS 2022_

### Official Review · Reviewer_fpV5 · 2022-10-05
**SADT combines SAM and self-distillation and outperforms them in experiments**

**Rating:** 1
**Confidence:** 2

**Review:**

This paper combines sharpness-aware minimization and self-distillation in one method called SADT. The presentation is reasonably clear. The experiments are impressive because SADT outperforms SAM, which is sota.  If the results are not cherry picked, this is a very important contribution -- which I would like to see as a follow up conference paper. Accept!

Please fix though: The legend in Figure 2 is much too small.

---

### Official Review · Reviewer_wD6Q · 2022-10-12

**Rating:** 1
**Confidence:** 4

**Review:**

This paper combines recent methods in improving NNs' generalization performance: SAM and self-distillation. The initial experiments results look good and the method is interesting. Thus, I vote for acceptance.

---

### Official Review · Reviewer_EqEf · 2022-10-16
**Accept: Combining sharpness-aware minimization with self-distillation seems to improve performance**

**Rating:** 1
**Confidence:** 2

**Review:**

The paper proposes to combine sharpness-aware minimization with self-distillation to speed up convergence and improve test performance. Since the empirical results indicate a potential improvement in performance, I recommend to accept the paper.

Some simple suggestions to improve the paper:
- Provide results averaged over multiple random seeds.
- Improve the readability of the figures and the table by increasing the font size.
- Improve the clarity of the method section (Section 3).

---

### Decision · Program_Chairs · 2022-10-20

Accept